# The Role of NK and T Cells in Endometriosis

**DOI:** 10.3390/ijms251810141

**Published:** 2024-09-21

**Authors:** José Lourenço Reis, Natacha Nurdine Rosa, Catarina Martins, Miguel Ângelo-Dias, Luís Miguel Borrego, Jorge Lima

**Affiliations:** 1Department of Obstetrics and Gynecology, Hospital da Luz Lisboa, Luz Saúde, 1500-650 Lisboa, Portugal; jose.mlreis@gmail.com; 2UCD School of Medicine, University College Dublin, D04 V1W8 Dublin, Ireland; natachagarciarosa@gmail.com; 3CHRC, NOVA Medical School, Faculdade de Ciências Médicas, Universidade NOVA de Lisboa, 1169-056 Lisboa, Portugal; catarina.martins@nms.unl.pt (C.M.); miguel.dias@nms.unl.pt (M.Â.-D.); borregolm@gmail.com (L.M.B.); 4Immunology Department, NOVA Medical School, Faculdade de Ciências Médicas, Universidade NOVA de Lisboa, 1169-056 Lisboa, Portugal; 5Department of Imunoallergy, Hospital da Luz Lisboa, Luz Saúde, 1500-650 Lisboa, Portugal

**Keywords:** endometriosis, NK cells, T cells, peritoneal fluid, peripheral blood

## Abstract

Endometriosis, a debilitating condition, affects one in ten women of reproductive age. Its pathophysiology remains unclear, though deficiencies in immune surveillance are thought to create an environment conducive to the evasion of ectopic endometrial cells from the immune system. Our research explores the immunological impact of endometriosis both locally and systemically, emphasizing natural killer (NK) and T cell subpopulations. We incorporated 62 female patients who underwent laparoscopic surgery; of those, 47 had endometriosis, and 15 were controls. We collected peritoneal fluid (PF) and peripheral blood (PB) samples which were tagged with monoclonal antibodies and subsequently scrutinized using flow cytometry. Our findings revealed significant differences in immunological profiles based on demographic factors and symptomatology. In the endometriosis cohort, there was an increase in PB CD56^Hi^CD16^dim^ and PF CD8^+^ CD56^dim^CD16^Hi^ NK cells. CD16^+^ CD4 T cell levels were significantly lower in the PB of endometriosis patients who smoke. Individuals with more severe disease displayed significantly higher levels of PB CD16^+^ CD8 T cells, which also increased in those with non-menstrual pelvic pain. Dysmenorrhea severity correlated with a progressive increase in PF CD8^+^ CD56^dim^CD16^Hi^ NK cells. These variations in specific lymphocyte subsets, namely, within NK and T cells, suggest potential immunological mechanisms in the evolution and clinical presentation of endometriosis.

## 1. Introduction

Endometriosis (EM) is a chronic gynecological condition characterized by the presence of endometrial tissue outside the uterine cavity, typically in the pelvic peritoneum, ovaries, and bowel. These ectopic lesions result in symptoms including dysmenorrhea, dyspareunia, pelvic discomfort, and infertility. Despite the fact that EM impacts one in ten women of reproductive age [1], the pathophysiology remains unclear, hence, effective diagnostic and treatment strategies are still yet to be established.

The most accepted theory regarding pathogenesis is that of retrograde menstruation [2]. However, though this process occurs in almost all women, only 10% develop EM [1], suggesting that additional factors must be present for EM to develop. The recent literature implicates immunological dysfunction in EM pathophysiology [3,4,5,6,7,8,9,10,11,12,13,14,15,16,17,18,19,20]. It is suggested that immune surveillance deficiencies in the peritoneal cavity create a conducive microenvironment for ectopic endometrial cell immune evasion and adherence. This environment promotes proliferation and angiogenesis, allowing for abnormal tissue survival and settlement [3,4,5,6,7,8,9,10,11,12,13,14,15,16,17,18,19,20]. Many studies suggest that changes in the immune profile, particularly in natural killer (NK) cells and T cells, underpin the compromised immune response observed in women with EM, resulting in poor clearance of ectopic endometrium [3,4,5,6,7,8,9,10,11,12,13,14,15,16,17,18,19,20]. To our knowledge, 18 studies [3,4,5,6,7,8,9,10,11,12,13,14,15,16,17,18,19,20] have investigated NK cell and T cell activity or phenotypic changes in the peripheral blood (PB) and peritoneal fluid (PF) of EM patients. According to the literature, NK cell cytotoxicity is diminished in EM, reflecting a functional defect. This defect is characterized by an overexpression of NK cell inhibitory receptors and a decreased expression of activating receptors [6,8,10,21,22]. For lymphocyte subpopulations, both quantitative and functional defects were observed [4,12,19], but the existing literature provides limited clarity. More specifically, Iwasaki et al. [19] reported an increase in suppressive T cells and a decrease in cytotoxic T cells in PB and PF samples of EM patients. Contrastingly, Hassa et al. [7] observed no significant changes in lymphocyte subsets in EM patients. More recently, however, Meggyes et al. [5] highlighted quantitative differences in specific lymphocyte subpopulations, including decreased T, CD4^+^ T, and regulatory T cells. They also noted functional differences, such as differential TIM3/Galectin-9 T cell expression [5].

Further research is needed to understand the immune aspect of EM, thus enabling the development of more targeted and effective treatments. This study intends to contribute insights to a field in which understanding is far from comprehensive. It distinguishes itself by examining the connection between immunology, demographic factors, and symptoms. To this end, we explore the immunological impact of EM both locally (PF) and systemically (PB), specifically focusing on NK and T cell subpopulations. We establish a baseline in control and case groups. Our approach offers a unique clinical perspective from previous studies in the literature, with noticeable methodological differences. In more detail, we identify abnormal patterns in NK and T cell profiles relating to clinical presentation and symptoms such as dysmenorrhea, dyspareunia, pelvic pain, rectorrhagia, and dyschezia. Moreover, we examine the interplay between immunology and demographic factors, like race and smoking status, on both systemic and local NK and T cell populations of control and EM patients. By contributing to a deeper understanding of the interaction between EM and immune and demographic factors, we hope to lay the groundwork for the creation of personalized treatments and interventions for individual clinical presentations. This approach may enhance treatment efficacy while potentially minimizing adverse or off-target effects.

## 2. Results

### 2.1. Demographic and Clinical Characteristics of Patients

The demographic and clinical data for both enrolled groups are summarized in Table 1. In the control group, the average age was 38.5 years, with a standard deviation (SD) of 9.2, and the age ranged from 20 to 55 years. In the EM group, the average age was 36.2 years, with an SD of 6.34, and the ages ranged from 31 to 42 years. Most participants in both groups were White (>85%). There were no significant differences in the demographic and clinical characteristics between the two groups, apart from the history of infertility, which was significantly more common in women with EM (*p* = 0.003). With regards to hormone therapy, the difference in its usage between 33% of the control group and 66% of the EM group was not statistically significant (*p* = 0.09).

### 2.2. Endometriosis Characteristics

Table 2 provides detailed information on the characteristics of EM in the participating patients. The average time between the EM diagnosis and surgery was 2.8 years. A majority of patients had severe forms of the disease (78.7%), corresponding to stages III/IV. The most frequently experienced symptom was dysmenorrhea, reported by 93.6% of participants, with severe dysmenorrhea (≥7) reported by 76.6%. Other symptoms included dyspareunia (59.6%), chronic pelvic pain (40.4%), dyschezia (34.0%), bowel symptoms (31.9%), urinary tract symptoms (10.6%), and rectal bleeding (8.5%). Furthermore, about 4.3% of cases involved extra-pelvic EM.

### 2.3. Tissue-Resident and Circulating Lymphoid Cells: Differences between Peripheral Blood and Peritoneal Fluid

As expected, the lymphoid cellular content differed significantly between PB (circulating) and PF (tissue-resident) samples. PF samples displayed a significantly larger proportion of lymphocytes within total leukocytes (*p* < 0.001), despite having lower absolute cell counts (when considering total lymphocytes, as well as T and NK cells).

Regarding T cells, the ratios of CD4/CD8 T cells were considerably higher in PB than in PF samples (*p* < 0.001). PB displayed increased percentages of CD4 T cells (representing 67% of total T cells, *p* < 0.001) and lower percentages of CD8 T cells (constituting 33% of total T cells, *p* < 0.001). In contrast, PF CD4 T cells accounted for only 37% of total T cells, whereas CD8 T cells were more abundant, with mean values of 63%. CD56^+^ T cells also varied between compartments, amounting to roughly 15% in PF compared to only 10% in PB (*p* < 0.001). Additionally, tissue-resident lymphocytes exhibited higher percentages of CD16 in both CD4 and CD56^+^ T cells (*p* < 0.001 for both) and higher percentages of CD56 in CD8 T cells when compared to circulating samples (*p* < 0.001).

Interestingly, no significant differences were observed in total NK cells between PB and PF samples. However, in PB, two major circulating NK subsets were identified: CD56^dim^CD16^Hi^, which was the most abundant, and the less represented CD56^Hi^CD16^dim^. In PF samples, four distinct subsets were found: CD56^+/dim^CD16^Hi^ and CD56^Hi^CD16^−^ NK cells were the most abundant, while CD56^Hi^CD16^+^ and CD56^dim^CD16^−^ NK cells were less common. It is worth noting that a larger percentage of circulating NK cells expressed CD8 in comparison to PF NK cells (*p* < 0.001).

The complete statistical data for the results discussed in this section can be found in Appendix A.

### 2.4. Patients with Endometriosis: Impact on Circulating and Local Immune Profiles

The next step in our analysis involved further identifying the differences in both sample types between EM patients and the control group.

Initially, only modest differences between controls and EM patients were observed in major circulating subsets. For example, there was a trend towards higher absolute T cell counts in the PB of EM patients (*p* = 0.054) and lower percentages of circulating NK cells (*p* = 0.081). Within the NK cells, despite being the most abundant subset, cytotoxic CD56^dim^CD16^Hi^ NK cells were slightly decreased in EM patients (*p* = 0.073). Conversely, there were significantly increased percentages (*p* = 0.019) and absolute counts (*p* = 0.047) of CD56^Hi^CD16^dim^ NK cells compared to controls. Therefore, EM patients showed lower CD56^dim^CD16^Hi^ NK cells/CD56^Hi^CD16^dim^ NK cells ratios than controls (*p* = 0.019).

The analysis of PF samples was, however, less effective in identifying differences between EM patients and controls. In fact, the only notable difference was the higher percentages of CD8^+^ CD16^Hi^CD56^dim^ NK cells observed in EM patients compared to controls (*p* = 0.029).

The complete statistical data for the results discussed in this section can be found in Appendix A

### 2.5. Race and Endometriosis: Impact on Circulating and Local Immune Profiles

Some of the reported differences were more pronounced in Black EM patients compared to White patients. According to Figure 1, Black EM patients demonstrated significantly higher percentages of total lymphocytes (*p* = 0.008), CD56^Hi^CD16^dim^ NK cells (*p* = 0.032), consequently leading to lower ratios of CD56^dim^CD16^Hi^ NK cells/CD56^Hi^CD16^dim^ NK cells (*p* = 0.025) in PB. Moreover, pertaining to peritoneal fluid, Black patients exhibited higher percentages of CD8^+^ CD56^dim^CD16^Hi^ NK cells (*p* = 0.033) and CD8^+^ CD56^Hi^CD16^dim^ NK cells (*p* = 0.009) compared to their White counterparts.

### 2.6. Age and BMI Effect on Endometriosis: Impact on Circulating and Local Immune Profiles

In contrast to race, age and body mass index (BMI) appeared to have no significant impact in the subsets studied. On the periphery, the most notable variation identified was that absolute counts of CD56^Hi^CD16^dim^ NK cells were higher in younger EM patients (age < 37.5 years) compared to a similar subgroup of control women (age < 37.5 years) (*p* = 0.020). Locally, the group of younger women with EM exhibited significantly higher percentages of CD56^+^ T cells (*p* = 0.017) than young controls.

### 2.7. Smoking and Endometriosis: Impact on the Circulating Immune Profile

Smoking habits seem to significantly affect the circulating immune profiles of patients with EM. As depicted in Figure 2, both circulating T cell and NK cell populations are impacted. For T cells, CD16^+^ CD4 T cells were markedly lower in EM patients who smoked compared to those who did not. The ratios of CD56^dim^CD16^Hi^ NK cells to CD56^Hi^CD16^dim^ NK cells, which have been previously reported to decrease in EM patients compared to controls, were even more conspicuously lower in EM patients who did not smoke. All individuals in the control group were non-smokers.

### 2.8. Extrauterine Locations of EM: Cause or Consequence of Altered Immune Profiles

A detailed characterization of the studied EM group allowed the assessment of local and systemic immune profiles based on the presence of extrauterine endometriotic tissue, specifically in the bowel. Women with bowel EM (identified during surgical intervention) demonstrated lower percentages of circulating CD56^+^ CD8 T cells (*p* = 0.033). Conversely, they exhibited higher percentages of CD56^Hi^CD16^dim^ NK cells (*p* = 0.038). This led to a noticeable decrease in the circulating CD56^dim^CD16^Hi^ NK cells/CD56^Hi^CD16^dim^ NK cells ratios (*p* = 0.039) in these women compared to the control group (Figure 3).

### 2.9. Hormonal Contraception and Endometriosis: Impact on Circulating Immune Profile

As observed in Figure 4, hormonal contraception interferes with CD8 expression in NK cells. Lower values of circulating CD8^+^ NK cells and CD8^+^ CD16^Hi^ NK cells were observed in the control group under therapy compared to all other groups, specifically with EM patients. However, their impact on EM remains unclear, as no differences were observed between EM with and without contraception. Furthermore, despite a slight decrease in both EM subgroups, the percentages of CD8^+^ NK cells were similar to those of controls without hormonal contraception.

### 2.10. Endometriosis Severity and Length of Exposure: Impact on Circulating and Local Immune Profiles

The grade of EM and the median years since diagnosis were also considered in the investigation of local and systemic immune profiles. Patients with lower EM grades (i.e., grades 1 and 2) and those diagnosed up to 2 years before enrollment exhibited increased percentages of PF CD8^+^ CD56^dim^CD16^Hi^ NK cells compared to controls (*p* = 0.033 for grade 1/2 patients; *p* = 0.008 for patients with <2 years of EM diagnosis). In addition, patients diagnosed less than 2 years prior showed higher percentages of CD16^+^ CD8 T cells (*p* = 0.027), while those diagnosed more than 2 years prior may have seen these values decrease to levels akin to controls.

Disease severity appears to impact peripheral cell populations as well. Figure 5 shows that EM patients with grades 1/2 begin with lower levels of circulating CD16^+^ CD8 T cells, which then progressively increase with disease severity (*p* = 0.031). However, CD8^+^ CD56^+^ T cell percentages remain comparable between controls and EM patients with grades 1/2. Interestingly, these values decrease in EM patients with increasing disease severity, specifically in grades 3 and 4 (*p* = 0.050). A similar pattern is observed in PF samples for CD8^+^ CD16^Hi^ NK cells. Despite decreases in EM grades 3 and 4, patients with grades 1/2 present higher percentages than controls (*p* = 0.033).

### 2.11. Clinical Features of EM: The Relationship between the Immune System and Symptomatology

Finally, we examined the relationship between discrete clinical features and specific symptomatology concerning the immune parameters evaluated in both peritoneal and circulating lymphoid compartments.

#### 2.11.1. Dysmenorrhea

Concerning dysmenorrhea (Figure 6), EM patients who did not have this complaint showed higher percentages of both CD16^+^ CD8 T cells and CD56^+^ T cells in their peritoneal fluid (*p* = 0.033, and *p* = 0.037, respectively). Furthermore, when dysmenorrhea was classified according to its severity grade, a rise in PF CD8^+^ CD56^dim^CD16^Hi^ NK cells was observed among patients with increased severity (*p* = 0.031).

#### 2.11.2. Dyspareunia

Furthermore, no significant differences were observed in circulating lymphoid cell populations when comparing patients based on complaints of dyspareunia. Interestingly, however, EM patients without symptoms of dyspareunia had increased levels of circulating CD56^Hi^CD16^dim^ NK cells (*p* = 0.034), thereby demonstrating significantly lower ratios of CD56^dim^CD16^Hi^ NK cells to CD56^Hi^CD16^dim^ NK cells (*p* = 0.036) compared to controls (Figure 6).

#### 2.11.3. Non-Menstrual Pelvic Pain

In patients with EM suffering from non-menstrual chronic pelvic pain, significant differences were recognized in circulating lymphocyte levels. More precisely, these patients showed higher CD4 and total T cell counts in comparison to controls (*p* = 0.015 and *p* = 0.029, respectively) while also displaying elevated percentages of CD16^+^ CD8 T cells in PF (*p* = 0.046) (Figure 6).

#### 2.11.4. Rectorrhagia

In the context of rectorrhagia, several differences were observed in immune parameters (Figure 7). Firstly, EM patients who reported rectorrhagia had significantly fewer circulating CD8^+^ NK cells than both controls and EM patients without rectorrhagia (*p* = 0.006). Interestingly, EM patients without rectorrhagia possessed even higher percentages of circulating CD8^+^ NK cells compared to controls.

Additionally, patients with rectorrhagia displayed increased levels of circulating CD56^Hi^CD16^dim^ NK cells (*p* = 0.046), indicating significantly lower circulating CD56^dim^CD16^Hi^ NK cells/CD56^Hi^CD16^dim^ NK cells ratios (*p* = 0.046) than controls. On a local level, the PF samples of EM patients with rectorrhagia contained significantly higher levels of CD56^+^ CD4 T cells compared to controls (*p* = 0.036). Once more, EM patients without rectorrhagia also differed from controls by showing a decrease in CD56^+^ CD4 T cells.

#### 2.11.5. Dyschezia

Finally, dyschezia, also known as obstructed defecation syndrome, was reported in some EM patients (34%). However, alterations in lymphoid populations seem to occur particularly in patients without this condition. As shown in Figure 8, EM patients without dyschezia exhibited increased CD4, CD8, and total T cell counts (*p* = 0.048; *p* = 0.019; and *p* = 0.016, respectively) in PB. Locally, PF analysis demonstrated lower CD56^+^ CD4 T cells in patients without dyschezia compared to both controls and EM patients with dyschezia (*p* = 0.005).

When examining dyschezia according to its severity, we noted that EM patients suffering from intense dyschezia demonstrated lower circulating CD8 T cell counts (*p* = 0.021). In contrast, patients reporting mild to moderate dyschezia exhibited higher total T cell (*p* = 0.016) and CD4 T cell counts (*p* = 0.043), along with lower NK cell percentages (*p* = 0.026). Similarly, in PF, patients with severe dyschezia also significantly showed lower percentages of CD8^+^ CD16^+^ CD56^+^ NK cells (*p* = 0.038).

## 3. Discussion

In this study, we investigated the effects of EM on both the systemic (PB) and local (PF) immune profiles of patients. We specifically focused on the variations of NK and T cell populations in comparison to a control group. To our knowledge, 18 studies [3,4,5,6,7,8,9,10,11,12,13,14,15,16,17,18,19,20] have already investigated NK cell and T cell activity or the expression of specific NK cell subsets in the PB and PF of EM patients. However, our study introduces a unique clinical perspective, significantly diverging from the methodologies and parameters previously examined. First, our research deviates from the current literature by establishing a baseline, comparing specific T and NK cell populations in PB versus PF of control cohorts. Additionally, we characterized patterns of aberrant NK and T cell profiles according to clinical presentation (dysmenorrhea, dyspareunia, non-menstrual pelvic pain, rectorrhagia, dyschezia), exploring the correlation between immune subsets and symptoms. Furthermore, we formally assessed the impact of demographic factors, such as race and smoking status, on systemic and local NK cell and T cell populations in control and EM patients. One notable divergence of our study is that the exclusion of EM in the control group was confirmed by laparoscopic surgery with direct visualization of the pelvis, thereby providing a fully reliable method for confirming or refuting the presence of EM rather than relying solely on clinical inquiry.

As expected, our comparative analysis revealed significant differences in the immune environments of PB and PF. A previous study [23] also investigated baseline immunological differences between PB and PF, but we chose to analyze additional NK and T cell subsets that were not covered in this study. Absolute cell counts, including total lymphocytes, were significantly lower in PF compared to PB. However, lymphocytes were present in a significantly greater proportion of total white cells in PF. PF also exhibited smaller CD4/CD8 T cell ratios, showing an increased abundance of CD8 T cells at the expense of CD4 T cells. Conversely, as expected, the opposite relationship was observed in PB, which was richer in CD4 T cells compared to CD8. This is in agreement with Guo et al. [23], who also found a relative abundance of CD8 T cells and increased global T cell activation in PF compared to PB via mass cytometry. The increased proportions of lymphocytes and lower CD4/CD8 ratios in PF may suggest enhanced cytotoxic responses in the peritoneal space, likely due to local immune adaptation relating to proximity to abdominal organs and possible exposure to gut microbial challenges. In essence, the peritoneal space seems to be a dynamic, pro-inflammatory immunological microenvironment [23,24], which is an important consideration when studying the pathogenesis of EM. Supporting this hypothesis, we found that CD16^+^ CD4 T cells, CD16^+^ CD56^+^ T cells, CD56^+^ T cells, and CD56^+^ T cells were proportionally more abundant in PF than PB.

Interestingly, total NK cells in our cohort did not significantly differ between PB and PF samples as compared to other studies [23]. Indeed, in the study by Guo et al. [23], the peritoneal compartment showed an increased NK cell presence compared to PB. Several causes can be noted for this discrepancy, such as the heterogeneity of PF introducing sample variation, the timing of sample collections, and the small sample size of our control group, which limits statistical power. However, a detailed analysis of NK cell subsets did reveal important differences. First, two predominant NK subsets were identified in PB (the abundantly present CD56^dim^CD16^Hi^ and less prevalent CD56^Hi^CD16^dim^), while PF samples showed four identifiable NK subsets (in order of abundance: CD56^+/dim^CD16^Hi^, CD56^Hi^CD16^−^, CD56^Hi^CD16^+^, and CD56^dim^CD16^−^ NK cells). Moreover, CD8^+^ NK cells were also increased in PB compared to PF. Our data align with the established literature [21,25,26], which states that CD56^Hi^CD16^dim^ NK cells are less cytotoxic and prominent cytokine producers and are abundantly found in secondary lymphoid tissues and among uterine cells [27]. The suggested mature NK cells CD56^dim^CD16^Hi^, with stronger cytotoxic capacities, are more abundant in PB, where protection against aggression requires more potent surveillants. These variations in local and systemic NK cell subsets may partially underpin the pathogenesis of EM, thus creating a local immune environment that promotes immune evasion, cytokine-mediated symptoms, and inflammatory changes [21].

After characterizing lymphoid subpopulations in PB and PF, we next assessed how the presence of EM disrupted immunological patterns. Hassa et al. [7] reported no significant differences in peritoneal major lymphoid populations (i.e., NK cells and CD4 and CD8 T cells) between controls and EM patients. Our results align with theirs for PF samples. However, our EM cohorts showed increased circulating CD56^Hi^CD16^dim^ NK cells and lower CD56^dim^CD16^Hi^/CD56^Hi^CD16^dim^ ratios, suggesting decreased NK cell cytotoxicity systemically compared to the general population. Additionally, our EM cohorts may be more susceptible to cytokine-mediated inflammation, leading to symptomatology, as corroborated by previous studies [20,28,29].

A more detailed analysis of the specific PF subsets allowed us to identify an increase in CD8^+^ CD56^dim^CD16^Hi^ NK cells in EM patients, which may be significant given that the CD8 molecule typically enhances NK cell cytotoxicity [30]. Although Schmitz et al. [31] found reduced perforin-positive CD8 T cells in the menstrual effluent of EM patients—indicating a reduced cytotoxic potential—other authors assert that CD8 T cells are increased in endometriotic lesions [32,33]. This somewhat mirrors our observations on CD8^+^ NK cell subsets.

Therefore, we hypothesize that while there may be an increase in CD8^+^ NK cells in EM PF—likely as an adaptive response to the disease—they might function as compensation for altered T cells. Alternatively, these cells could carry alterations in their cytotoxic potential that allow the development and establishment of endometriotic lesions. However, these variations in local CD8^+^ cells might be a consequence, not a cause of EM, according to D’Hooghe et al. [32]. Therefore, we contend that the role of CD8^+^ lymphocytes in EM remains controversial and necessitates further investigation [33].

### 3.1. Endometriosis: Immunology of Demographic and Disease Factors

To further understand the immunological landscape of EM, we analyzed differences across various demographic parameters. Race appeared to influence EM’s immunological profile. The differences identified, specifically higher percentages of circulating CD56^Hi^CD16^dim^ NK cells and lower ratios of circulating CD56^dim^CD16^Hi^ NK cells to CD56^Hi^CD16^dim^ NK cells, along with a local increase in CD8^+^ CD56^dim^CD16^Hi^ NK cells, were more pronounced in Black EM patients. In addition, Black EM patients exhibited higher overall PF lymphocyte and CD8^+^ CD56^Hi^CD16^dim^ NK cell absolute counts.

This phenomenon may be due to several reasons. A meta-analysis highlights a troubling trend: Black patients with EM are 50% less likely to receive an accurate diagnosis compared to their White counterparts [34]. Coupled with the reported average duration of 5-10 years for an EM diagnosis [35], this suggests Black EM patients may have more severe, late-stage disease, contributing to heightened disparity in medical, societal, and socioeconomic factors, and access to treatment or healthcare [34].

Genetic variation and predisposition might also influence EM pathogenesis. Indeed, certain genetic polymorphisms have been linked to an increased risk of EM [21,36,37], although these have predominantly been studied in Asian populations. Further investigation is necessary to confirm these associations in Black populations.

Another demographic parameter assessed was age, which apparently had a lower impact than race/ethnicity. Indeed, differences were primarily observed when comparing younger cohorts (<37.5 years) (EM vs. controls). These cohorts showed greater absolute counts of circulating CD56^Hi^CD16^dim^ NK cells and CD56^+^ T cell percentages in PF. The absence of significant differences in older cohorts aligns with the existing literature [38]. Thus, we hypothesize that as EM is an estrogen-dependent condition [38], disease severity and aberrant immune features are more pronounced during the hormonal and menstrual cycle and are alleviated by age-related estrogen decline and menopause. Interestingly, older EM patients (>37.5 years) exhibited higher percentages of CD8^+^ CD56^dim^CD16^Hi^ NK cells in PF compared to younger controls, aligning with previous studies reporting increased CD56^dim^CD16^Hi^ NK cells with aging [39,40]. This may be an adaptive function of the reproductive tract against possible pathogens, but its role in EM calls for further study.

Smoking status also affected the circulating immune profiles of EM patients. CD16^+^ CD4 T cells were significantly lower in EM patients who smoked compared to non-smokers. This suggests that smoking may impair T cell activation, promoting both immunosuppression and immune evasion in these patients, although the literature on this topic is controversial. Martos et al. found additional senescence-associated genes in CD4 T cells from smokers, leading to immune cell dysregulation [41], while Chapron et al. [42] found that smoking habits had no significant impact on EM risk or American Fertility Society stage/score. However, one significant limitation of our study is that all individuals in the control group were non-smokers; this precluded establishing a comprehensive baseline for comparison, making it difficult to discern the independent immunological effects of smoking, EM, and their relationship.

Women with extrauterine (intestinal) EM, often deemed the most aggressive form of EM, typically exhibit immune profiles similar to those lacking intestinal endometriotic lesions. Nonetheless, as confirmed earlier, patients with extrauterine EM reveal elevated percentages of CD56^Hi^CD16^dim^ NK cells and reduced CD56^dim^CD16^Hi^ NK cells/CD56^Hi^CD16^dim^ NK cells compared to the control group. This hints at a pro-inflammatory environment, conducive to processes like angiogenesis, pivotal in establishing ectopic endometriotic lesions [43]. Also, a significant decline in circulating CD8^+^ CD56^+^ T cells suggests a systemic reduction in immunosurveillance and lesion apoptosis. It is crucial to discern whether these immune modifications are a cause or an effect of a modified immune profile. As far as we know, no other studies have immunologically characterized extrauterine EM, with most of the literature reviewing changes based on severity instead of lesion location. Despite this, considering current knowledge and assuming our moderate trends are valid, we propose that women with extra-pelvic EM exhibit a more distinct immune and inflammatory microenvironment that systemically encourages processes, such as angiogenesis, thus helping establish ectopic endometriotic lesions at more remote locations [43]. This, however, needs more exploration, as present study limitations, particularly the small number of participants, could obscure additional insights.

Hormonal contraceptives have long been recognized as a standard treatment for EM, primarily aimed at providing symptomatic relief due to the condition’s dependence on estrogen. In our investigation, we evaluated whether hormonal contraception, a standard therapeutic approach, modifies immune profiles. Hormonal contraception reduced total peripheral lymphocyte levels in both control and EM groups. The literary findings on this topic are elusive. Though several studies have tried to characterize hormone-mediated immunological changes, they reported varied results based on the type of contraception, active ingredient, and lymphocyte subpopulation studied [44,45,46]. However, it is well-established that hormonal contraception is linked to changes in the frequency of immune cells in the genital mucosa [47] and a reduction in cytotoxicity and cytokine production [44]. Thus, this decrease in total lymphocyte count may substantiate the symptomatic relief provided by hormonal contraception owing to reduced cytokine production and inflammation. Upon investigating other subsets, hormonal contraception resulted in decreased counts of CD8-positive CD56^+^ T cells, CD56^dim^CD16^Hi^ NK cells, and total NK cells in the control group, but not in EM patients, indicating a relatively lower impact on the immune system in the latter group. Notably, Waiyaput et al. [48] observed that the combined oral contraceptive (COC), containing ethinyl estradiol and desogestrel, significantly reduced the macrophage cell count while enhancing NK and regulatory T cell counts in women with ovarian EM [48]. Prathoomthong et al. [49] also noted that dienogest increased NK cell counts in the eutopic endometrium. However, it is crucial to highlight that these observations were identified within the context of adenomyosis, not EM [49]. These variations could result from several factors including methodological differences, type and duration of contraceptive use, distinct hormonal components, and individual hormonal level fluctuations. Consequently, future research should employ a more targeted and comprehensive approach to decipher the immunological impacts of hormonal contraception on EM.

Recently diagnosed EM (less than 2 years) appear to display more noticeable immune alterations, such as increases in CD16^+^ CD8 T cells and CD8^+^ CD56^dim^CD16^Hi^ NK cells. This pattern stabilizes 2 years after diagnosis, leading to speculation about potential immune exhaustion. Several studies support this hypothesis, with cohorts with EM showing reduced T cell and NK cell effector function, suggesting a failure in immunosurveillance [3,50,51].

When assessing immune profiles based on EM severity, patients with EM showed progressively lower percentages of circulating CD56^+^ CD8 T cells with increasing disease severity. This decrease supports the hypothesis that ectopic endometrial cells evade the immune system. However, the literature presents conflicting viewpoints, with some studies suggesting a decrease in T cell populations in EM [12,17], while others report no significant differences [9,33,52], or even an increase [53,54]. It is pivotal to determine whether this decrease is a causative factor, indicating a predisposition to more severe disease, or whether a persistently chronic inflammatory microenvironment induces apoptosis and thereby causes a decline in these populations. Notably, the percentages of CD16^+^ CD8 T cells in PB were lower in stages 1, 2, and 3 of EM compared to controls. In stark contrast, individuals with more severe disease (stage 4) exhibited an increase in CD16^+^ CD8 T cells compared to controls and other EM groups. One possible explanation could relate to the dynamic nature of the immune response at various stages of EM. The evolving pathology might interact with the immune system, causing fluctuations in T cell populations. The functional role of CD16^+^ CD8 T cells in severe disease remains unclear. They could signal a stronger or more specific immune response in serious cases. Conversely, they could be a byproduct of disease progression in response to a chronic inflammatory microenvironment. As previously mentioned, increased cell counts do not automatically mean an enhanced immune response or cytotoxicity since immune exhaustion and inhibitory mechanisms need to be taken into account [50,51]. The body of evidence on this topic is much more conflicted and ambiguous [9,12,17,33,52], which underscores the need for further research considering potential confounding variables. This factor was noted by Slabe and colleagues, who reported fluctuating regulatory T cell concentrations in response to altered immune responses, such as increased serum cortisol [52].

PF analysis revealed an increased percentage of CD8^+^ CD56^dim^CD16^Hi^ NK cells in EM cohorts compared to controls. Notably, this increase was more pronounced in minimal/mild disease (stages 1 and 2), suggesting a heightened immune response in these patients, potentially reflecting the immune system’s effort to control the disease process. Hassa et al. reported no significant changes in PF NK cell populations between control groups and early- and late-stage EM, despite the study’s focus on the NK compartment, which was as in-depth as ours [7]. Thus, our study underscores the importance of extended immune characterizations to better understand the EM mechanisms.

Overall, our evaluation of immune profiles pertaining to EM progression reveals a complex interplay within the immune system. This contributes to the complexity of the existing literature on the subject matter that has yet to be entirely deciphered.

### 3.2. Endometriosis: The Relationship between Symptomatology and Immunological Profiles

To encompass the clinical aspects of EM, we investigated the relationship between immune profiles and symptomatic presentations, considering factors like dysmenorrhea, dyspareunia, non-menstrual pelvic pain, rectorrhagia, and dyschezia.

Dysmenorrhea is a prevalent symptom of EM, apparently demonstrating a complex interplay between immune cell populations and their pathogenesis. Notably, higher percentages of PF CD16^+^ CD8 T cells and CD56^+^ T cells were observed in patients without dysmenorrhea. Although there are no studies regarding EM-associated dysmenorrhea, other investigations have implicated CD8^+^ T cells in pain suppressive mechanisms. Increased levels of CD8^+^ T cells were found in sural nerve biopsies of murine subjects with diabetic peripheral neuropathy [55,56] and in the cerebrospinal fluid (CSF) of HIV patients with peripheral neuropathy [57]. This suggests that CD8^+^ T cells (and potentially other T cell subpopulations) provide a pain-suppressing or protective immunophenotype. However, it is crucial to note that increased CD8^+^ T cells have also been implicated in acute and chronic pain processes, such as Herpes Zoster neuralgia [56,58]. This dual role of CD8^+^ T cells in both pain suppression and exacerbation in different pain conditions illuminates the elusive nature of their role in pain modulation and emphasizes the need for future research on their relationships with dysmenorrhea.

The pathophysiology of dyspareunia is multifactorial, involving structural (e.g., pelvic anatomy), psychosocial, and inflammatory factors [59]. EM-associated dyspareunia is suggested to be due to the presence of endometriotic lesions in the pouch of Douglas [60]. For the immune profile of these patients, our results suggest that dyspareunia in EM might not be associated with systemic immune alterations. On the contrary, EM patients without reported dyspareunia displayed elevated levels of circulating CD56^Hi^CD16^dim^ NK cells and lower ratios of circulating CD56^dim^CD16^Hi^/CD56^Hi^CD16^dim^ NK cells compared to controls. Noticeably, a study by Guevoghlanian-Silva et al. found increased Transforming Growth Factor Beta (TGF) expression in patients with EM who reported dyspareunia [60]. TGF is known to modulate immune responses, limiting NK cell expansion/proliferation by keeping them in a quiescent state [61,62], potentially modulating the predominance of some NK subsets, as we report here.

Our results indicate that EM patients with non-menstrual pelvic pain exhibit elevated circulating T cell counts, including both CD4 and CD8 T cell subsets. This increased presence of T cells suggests an active adaptive immune response, potentially indicative of an ongoing inflammatory process. Moreover, PF analysis of EM patients with pelvic pain characterized higher percentages of CD16^+^ CD8 T cells. This could imply the existence of a local cytotoxic response, either in response to painful stimuli or causing tissue damage and inflammation, thereby resulting in pelvic pain.

To our knowledge, the immunopathogenesis of pelvic pain in EM has only been explored in one previous study conducted by He et al. [63]. Our results partially support their findings, particularly in recognizing the association between pelvic pain and systemic immune dysregulation, but there are significant distinctions. In fact, while He et al. [63] reported higher circulating NK cells, our study only demonstrated differences in the absolute counts of T cells, which were higher in patients with this complaint. This variation in T cell count is supported by the broader literature, whereby regulatory T cell dysfunction has been associated with the exacerbation and severity of EM [53,54].

Rectal bleeding associated with EM is attributed to the implantation of deep ectopic endometrial lesions on the rectal wall. This manifestation of EM is sometimes referred to as Digestive System Infiltrating Endometriosis (DSIE) [64]. Given its relatively low incidence [64], the literature is scarce on this topic, which is why we aimed to clarify the immunological landscape of rectal bleeding.

Systematically, EM patients with rectal bleeding showed lower percentages of CD8^+^ NK cells in PB compared to those without rectal bleeding and controls. These results suggest a potential immunoregulatory response or pathogenic role of NK cells in the context of rectal bleeding. Reduced NK activity in DSIE has also been observed in other studies [60,64,65], whereby an enhanced release of cytokines in EM impedes NK cell proliferation and activity [60,64,65]. Additionally, we observed that EM patients without rectal bleeding displayed a significantly larger proportion of CD8^+^ NK cells compared to controls. A further analysis of PB revealed that EM patients with rectal bleeding had higher levels of CD56^Hi^CD16^dim^ NK cells along with a significantly lower ratio of circulating CD56^dim^CD16^Hi^/CD56^Hi^CD16^dim^ NK cells compared to controls and EM patients without rectal bleeding. These findings further support the hypothesis of an immune microenvironment characterized by pro-inflammatory cells and reduced cytotoxicity, allowing immune evasion and the establishment of endometriotic lesions on the rectal wall.

Locally, EM patients with rectorrhagia displayed significantly higher levels of CD56^+^ CD4 T cells in PF compared to controls. Conversely, EM patients without rectorrhagia showed a decrease in these cells compared to controls. These results further underscore the presence of a distinct immune environment associated with the gastrointestinal symptoms of EM. We hypothesize that the CD56^+^ CD4 T cell population increases in response to rectal bleeding, thereby managing inflammation linked to gastrointestinal symptoms. These findings align with the current literature, which notes an elevation in T cell populations, specifically regulatory T cells, in DSIE [64,66,67].

It is important to acknowledge the limitations of our study regarding the investigation of rectorrhagia. A confounding factor emerges, as rectal bleeding primarily derives from gastrointestinal pathologies [68], like hemorrhoids, rather than EM. Furthermore, this study encounters limitations in statistical power due to the exceedingly small sample size of EM patients with rectorrhagia enrolled.

Dyschezia, characterized by difficulty and pain during bowel movements, is a prevalent clinical presentation associated with bowel EM. When stratified by severity, it was observed that patients with severe dyschezia had lower circulating CD8 T cell counts, possibly indicating a compromised cytotoxic T cell response, which is crucial for immune surveillance of EM. Locally, patients with severe dyschezia exhibited reduced levels of CD16^+^ CD56^+^ CD8 NK cells, possibly pointing to potential NK cell exhaustion or diminished cytotoxic activity [3,50,51]. In simpler terms, prolonged or severe gastrointestinal symptoms may be contributing to immune exhaustion, affecting NK cell cytotoxicity. This variation could also result from increased expression of NK inhibitory receptors in severe EM, leading to localized immunosuppression [21].

The current literature concerning the immunological profile of bowel symptoms in EM is notably limited. However, the scarce available literature seems to contradict our research findings. According to Gueuvoghanian-Silva et al., IL-15 expression in deep rectosigmoid lesions was significantly greater in EM patients with dyschezia compared to those without dyschezia [60]. IL-15 is known to enhance CD8 T cell and NK cell expansion and survival [69,70] which, in turn, is expected to result in increased levels of these subsets. We observed the contrary. Though heightened IL-15 levels should theoretically contribute to an increase in NK cell and T cell populations, other studies have not explicitly examined this aspect, overlooking potential interactions with other cytokines and the broader immune environment. Overall, our study suggests a complex interplay between the immune system and gastrointestinal symptoms in EM, highlighting potential NK cell protective/regulatory mechanisms and distinct T cell immune profiles.

The literature regarding the relationship between lymphocyte subpopulations and symptoms such as dysmenorrhea, dyspareunia, or pelvic pain remain scarce. Moreover, it remains imperative to determine whether these deviations in the immune profiles are causes or consequences of EM and whether the findings are supported by studies in larger cohorts, such as those with more severe presentations. Our research disclosed significant differences in the immunological profiles of EM, contributing new insights to an area that currently lacks in-depth exploration, especially concerning demographic factors and symptomatology. Variations in particular lymphocyte subsets, such as NK cells and T cells, indicate the existence of potential protective immunological mechanisms, particularly concerning gastrointestinal and pelvic symptoms. This knowledge could enable the development of symptom-centric targeted therapies, addressing a crucial gap in current treatment methods. Another strength of our study is its thorough examination of specific subsets within NK and T cell populations, enhancing understanding of the immunological landscape of EM on a cellular level, which may hold key implications for its pathophysiology.

However, these strengths also illuminate some limitations of this study. While our research efficiently detailed the immunological profile or composition of PB and PF, it revealed a complex interplay among each subpopulation. Our study highlighted the presence of diverse immune NK and T cell subpopulations across various demographic and clinical presentations. However, it is vital to note that presence does not necessarily equate to activity. To heighten the breadth of immunological understanding, future research could include functional tests, such as cytotoxicity assays, cytokine analysis, and surface receptor analysis, to supplement these findings. Additionally, even though our sample population was generally larger than those in prior investigations of this topic, it is still relatively small for drawing definitive conclusions. This reduced statistical power is particularly evident in the examination of smoking (no individuals in the control group were smokers), dysmenorrhea, and rectorrhagia, with some groups having less than five enrolled patients in some instances. Furthermore, the changes observed in Black women with endometriosis should be interpreted with caution, given the absence of Black controls, limiting a more comprehensive comparison.

Future research should aim to clarify whether variations in immunological profiles correspond to changes in cellular activity. It will also be crucial to investigate the potential impact of genetic and hormonal influences on these profiles, as well as understand the functional interplay of different cell subsets. Nevertheless, our study paves the way for promising further research. The identification of specific immune cell subsets associated with demographic factors and symptomatology boosts our understanding of EM pathogenesis and promotes the development of more personalized and targeted treatments. By tailoring interventions based on individual clinical presentations, we can potentially amplify treatment efficacy and reduce adverse effects [71]. Notably, the observed variations in NK and T cell populations call for consideration of implementing NK and T cell therapies, similar to those used in cancer therapy, in the field of EM [72].

## 4. Materials and Methods

### 4.1. Study Design and Participants

This cross-sectional study, conducted from June 2021 to December 2022, involved 62 female patients, 47 of whom were diagnosed with EM, and 15 served as controls, without the disease. All participants were adult women of reproductive age treated at Hospital da Luz Lisboa and Hospital Beatriz Ângelo’s outpatient clinics. The EM group underwent laparoscopic surgery for the condition, while the control group had surgery for other pathologies, all under general anesthesia. Diagnosis of EM initially relied on clinical and imaging findings, later confirmed histologically according to the Human Reproduction and Embryology (ESHRE) guidelines [73]. All the patients included in the endometriosis group had histopathological confirmation of the disease through material collected during laparoscopic surgery.

The exclusion criteria encompassed women who had been pregnant within the past year, as well as those with immunological, endocrine, neoplastic, or other chronic diseases. Individuals suspected of complications due to pelvic inflammatory disease were also excluded. This study received approval from the Ethics Committees of Beatriz Ângelo Hospital, Hospital da Luz Lisboa, and NOVA Medical School. We ensured all patients were comprehensively informed about the research and the use of their biological material for scientific purposes. All study participants provided written informed consent before their involvement in this study.

### 4.2. Demographic and Clinical Data

Data collected for all patients encompassed age, race, BMI, smoking status, hormone therapy, and the time elapsed between the diagnosis of EM and surgery. Clinical data were gathered to assess disease severity, including symptoms such as dysmenorrhea, dyspareunia, chronic pelvic pain, dyschezia, bowel issues (constipation, peri-menstrual diarrhea, and bloating), urinary tract symptoms (hematuria and dysuria), rectorrhagia, and the presence of extra-pelvic EM. Disease severity was classified based on the guidelines of the American Society for Reproductive Medicine [74].

### 4.3. Sample Collection

PB and PF samples were gathered from all patients during surgery. To lower the potential for blood contamination, PF was collected at the start of the procedure. A laparoscopic needle was used for fluid extraction from the pouch of Douglas. The samples were preserved in EDTA-coated tubes kept at room temperature following surgery and delivered to the Immunology laboratory at NOVA Medical School. All PF samples underwent processing on the day of their collection, while blood samples were dealt with within 24 h based on laboratory availability.

### 4.4. Flow Cytometry Immunophenotyping Analysis

Absolute counts of each cell subset were determined using a single-platform method with Trucount^TM^ tubes (BD Biosciences, San Jose, CA, USA) for both samples. Specifically, we incubated a standardized sample volume with CD45 in a Trucount^TM^ tube for 15 min, lysed it using a BD FACS lysing solution (BD Biosciences, San Jose, CA, USA), and collected the data. Afterward, we stained the PB and PF samples with a panel of monoclonal antibodies, including CD45, CD3, CD8, CD16, CD56, and CD57. A comprehensive list of the antibodies utilized is provided in the Appendix A.

PF samples were centrifuged before immunophenotyping to concentrate the cells. A lyse-wash protocol was applied using a BD FACS lysing solution (BD Biosciences, San Jose, CA, USA). All samples were processed using a BD FACS Canto II (BD Biosciences, San Jose, CA, USA). Data analysis was conducted using Infinicyt™ (Cytognos, Salamanca, Spain) and FlowJo™ (BD Life Sciences, San Jose, CA, USA) software, both acquired from BD Biosciences (BD Life Sciences, San Jose, CA, USA). The gating strategy is illustrated in Figure 9.

### 4.5. Statistical Analysis

The demographic, clinical, and immunophenotypic data of the study participants were summarized using descriptive statistics. Categorical variables were reported as absolute and relative frequencies. Fisher’s exact test was used for comparisons between two groups when contingency tables had over 20% of expected values less than 5; otherwise, the chi-squared test was used. Continuous variables were displayed as mean (SD) (minimum-maximum) or as median and first-to-third quartile (Q1–Q3), depending on the context. In the Appendix A, all continuous variables were represented as mean (SD) along with a 95% confidence interval [95% CI], as well as the median and Q1–Q3. The 95% CIs were computed using Student’s t-distribution approximation for normally distributed continuous variables and bootstrap resampling for variables not normally distributed. Comparisons between two independent groups of normally distributed continuous data used Student’s t-test, while comparisons involving two dependent groups applied the paired-samples t-test. The Wilcoxon rank-sum test and the Wilcoxon signed-rank test were used for non-parametric comparisons. For comparisons involving more than two groups, ANOVA with post hoc Tukey’s HSD (honestly significant difference) test was used for normally distributed data, whereas the Kruskal–Wallis test with a post hoc Dunn’s test was utilized for data not normally distributed. The Shapiro–Wilk test assessed the normality of distributions. As this study was exploratory, no formal sample size calculation was conducted. Statistical significance was defined as *p* < 0.05. All analyses were performed using SAS software (version 9.4).

## 5. Conclusions

Previous studies have suggested that deficiencies in immune surveillance create a favorable environment for ectopic endometrial cells to evade the immune system. Our study investigated the immunological impact of EM both locally and systemically, focusing on NK and T cell subpopulations. Our research revealed significant differences in the immunological profiles of EM, providing new insights into a field that currently lacks comprehensive investigation, particularly concerning demographic factors and symptomatology. Identifying specific immune cell subsets linked to these factors enhances our understanding of EM pathogenesis and potentiates the development of more personalized and targeted treatments.

Future research should clarify if variations in immunological profiles correspond to alterations in cellular activity. It should also explore the functional interplay between these factors and the potential impact of genetic and hormonal influences on these profiles.

## Figures and Tables

**Figure 1 ijms-25-10141-f001:**
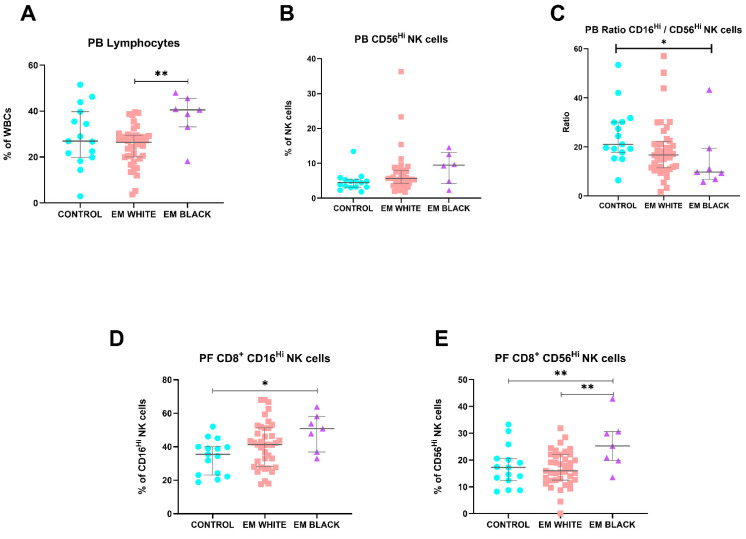
Differences in lymphocytes and lymphocyte subsets of PB and PF samples between healthy controls and endometriosis (EM) patients divided according to race. (**A**–**C**) Peripheral blood subsets: total lymphocytes, CD56^Hi^ NK cells, and CD16^Hi^ NK cells/CD56^Hi^ NK cells ratios, respectively. (**D**,**E**) Peritoneal fluid CD8^+^ CD16^Hi^ NK cells and CD8^+^ CD56^Hi^ NK cells. Graphs are presented as scatter dot plots with lines referring to medians and interquartile ranges. *p*-values obtained from KW—Kruskal–Wallis statistical analysis followed by multiple comparison tests. EM, endometriosis; PB, peripheral blood; PF, peritoneal fluid. * *p* < 0.05, ** *p* < 0.01.

**Figure 2 ijms-25-10141-f002:**
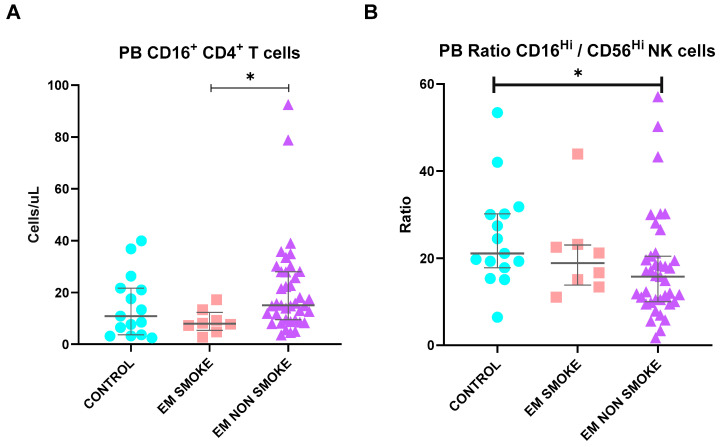
Differences in lymphocyte subsets of PB between healthy controls and endometriosis (EM) patients divided according to smoking habits, namely, peripheral blood (**A**) CD16^+^ CD4 T cells and (**B**) CD16^Hi^ NK cells/CD56^Hi^ NK cells ratios. Graphs are presented as scatter dot plots with lines referring to medians and interquartile ranges. *p*-values obtained from KW—Kruskal–Wallis statistical analysis followed by multiple comparison tests. EM, endometriosis; PB, peripheral blood; PF, peritoneal fluid. * *p* < 0.05.

**Figure 3 ijms-25-10141-f003:**
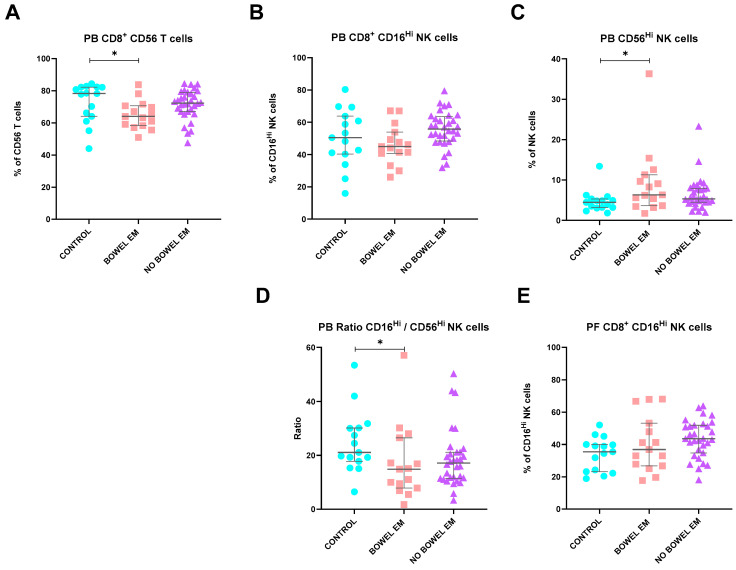
Differences in lymphocyte subsets of PB and PF between healthy controls and endometriosis (EM) patients divided according to the presence of bowel EM. (**A**–**D**) Peripheral blood subsets, CD8^+^ CD56^+^ T cells, CD8^+^ CD16^Hi^ NK cells, CD56^Hi^ NK cells, and CD16^Hi^ NK cells/CD56^Hi^ NK cells ratios. (**E**) Peritoneal fluid CD8^+^ CD16^Hi^ NK cells. Graphs are presented as scatter dot plots with lines referring to medians and interquartile ranges. *p*-values obtained from KW—Kruskal–Wallis statistical analysis followed by multiple comparison tests. EM, endometriosis; PB, peripheral blood; PF, peritoneal fluid. * *p* < 0.05.

**Figure 4 ijms-25-10141-f004:**
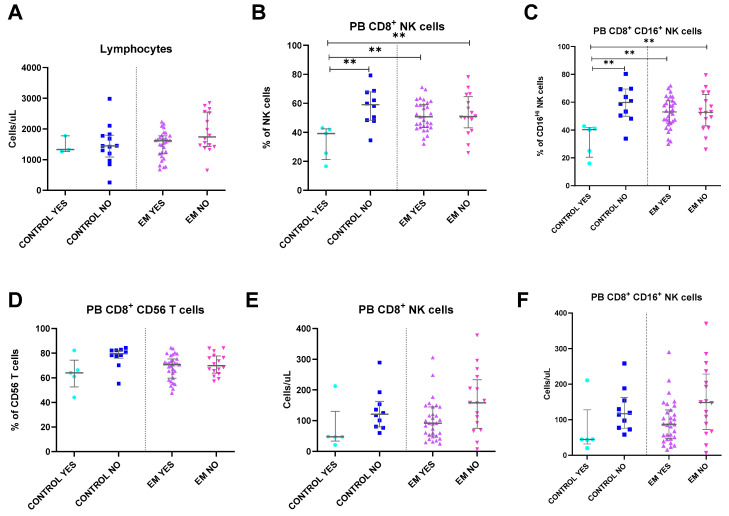
Differences in peripheral blood lymphocyte subsets between healthy controls and endometriosis (EM) patients divided according to the use of hormonal therapy. (**A**–**F**) Peripheral blood subsets, total lymphocytes, CD8^+^ CD56^+^ T cells, CD8^+^ NK cells, and CD8^+^ CD16^Hi^ NK cells. Graphs are presented as scatter dot plots with lines referring to medians and interquartile ranges. *p*-values obtained from KW—Kruskal–Wallis statistical analysis followed by multiple comparison tests. EM, endometriosis; PB, peripheral blood; PF, peritoneal fluid. YES, use of hormonal therapy. NO, no use of hormonal therapy. ** *p* < 0.01.

**Figure 5 ijms-25-10141-f005:**
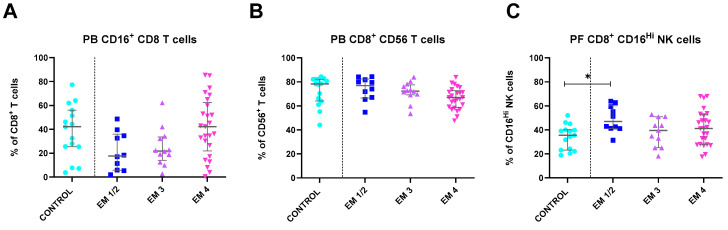
Differences in lymphocyte subsets of PB and PF between healthy controls and endometriosis (EM) patients divided according to disease severity. (**A**,**B**) Peripheral blood subsets, CD16^+^ CD8 T cells, and CD8^+^ CD56 T cells. (**C**) Peritoneal fluid CD8^+^ CD16^Hi^ NK cells. Graphs are presented as scatter dot plots with lines referring to medians and interquartile ranges. *p*-values obtained from KW—Kruskal–Wallis statistical analysis followed by multiple comparison tests. EM, endometriosis; PB, peripheral blood; PF, peritoneal fluid. * *p* < 0.05.

**Figure 6 ijms-25-10141-f006:**
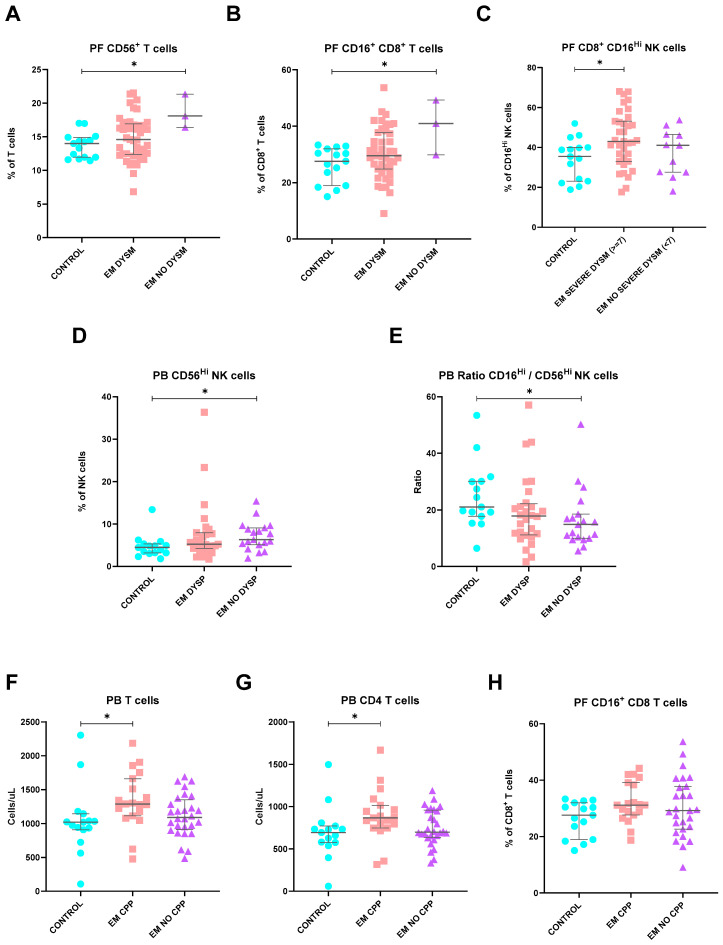
Differences in lymphocyte subsets of PB and PF between healthy controls and endometriosis (EM) patients divided according to the presence and severity of dysmenorrhea (**A**–**C**), dyspareunia (**D**,**E**), and chronic pelvic pain (CPP) symptoms (**F**–**H**). Graphs are presented as scatter dot plots with lines referring to medians and interquartile ranges. *p*-values obtained from KW—Kruskal–Wallis statistical analysis followed by multiple comparison tests. CPP, chronic pelvic pain; DYSM, dysmenorrhea; DYSP, dyspareunia; EM, endometriosis; PB, peripheral blood; PF, peritoneal fluid. * *p* < 0.05.

**Figure 7 ijms-25-10141-f007:**
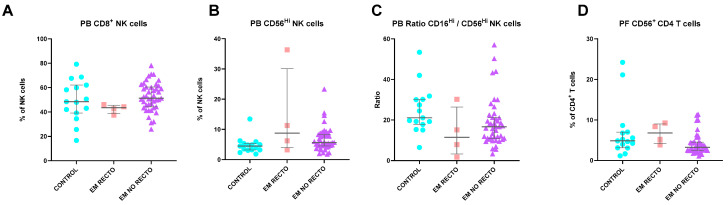
Differences in lymphocyte subsets of PB and PF between healthy controls and endometriosis (EM) patients divided according to the presence of rectorrhagia. (**A**–**C**) Peripheral blood subsets, CD8^+^ NK cells, CD56^Hi^ NK cells, and CD16^Hi^ NK cells/CD56^Hi^ NK cells ratios. (**D**) Peritoneal fluid CD56^+^ CD4 T cells. Graphs are presented as scatter dot plots with lines referring to medians and interquartile ranges. *p*-values obtained from KW—Kruskal–Wallis statistical analysis followed by multiple comparison tests. EM, endometriosis; PB, peripheral blood; PF, peritoneal fluid; RECTO, rectorrhagia.

**Figure 8 ijms-25-10141-f008:**
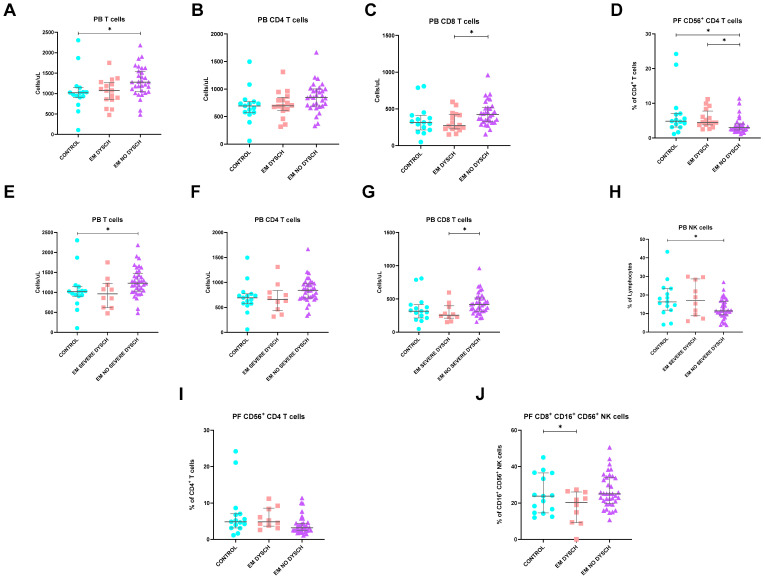
Differences in lymphocyte subsets of PB and PF between healthy controls and endometriosis (EM) patients divided according to the presence and severity of dyschezia symptoms. (**A**–**C**) Peripheral blood subsets with significant differences according to the presence of dyschezia, absolute counts of T cells, CD4 T cells, and CD8 T cells. (**D**) Peritoneal fluid CD56^+^ CD4 T cells according to the presence of dyschezia. (**E**–**H**) Peripheral blood subsets with significant differences according to the severity of dyschezia, absolute counts of T cells, CD4 T cells, CD8 T cells, and NK cell percentages. (**I**,**J**) Peritoneal fluid CD56^+^ CD4 T cells and CD8^+^ CD16^+^ CD56^+^ NK cells according to the severity of dyschezia. Graphs are presented as scatter dot plots with lines referring to medians and interquartile ranges. *p*-values obtained from KW—Kruskal–Wallis statistical analysis followed by multiple comparison tests. DYSCH, dyschezia; EM, endometriosis; PB, peripheral blood; PF, peritoneal fluid. * *p* < 0.05.

**Figure 9 ijms-25-10141-f009:**
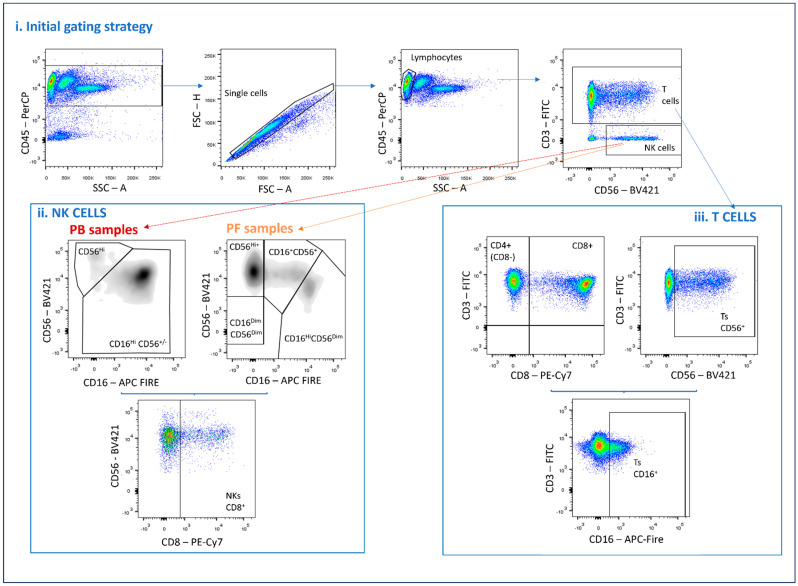
Gating strategy for lymphocyte subset characterization and marker expression in PB and PF samples. The maturation profile of NK cells was assessed according to the expression of NKG2A, KIR2DL1/CD158a, and CD57 as described elsewhere [75]. Fluorescence minus one (FMO)/fluorescence minus X (FMX) control strategies were used to properly set the threshold for positive staining for each marker in the respective populations of interest. All analyses were performed with FlowJo^TM^ Software, version X 10.0.7.

**Table 1 ijms-25-10141-t001:** Demographic characteristics.

	Control ^a^ (*n* = 15)	Endometriosis ^a^ (*n* = 47)	*p*-Value ^b^
**Age, years**			0.434
Mean (SD)	38.8 (9.2)	36.2 (6.34)	
Min, Max	20, 55	31, 42	
**Race, *n* (%)**			0.113
White	15 (100)	40 (85.1)	
Black	0 (0)	7 (14.9)	
**BMI, kg/m^2^**			0.430
Mean (SD)	23.2 (2.97)	25.2 (5.60)	
Min, Max	17.28	20.8, 28.0	
**Fertile, *n* (%)**			**0.003**
Yes	14 (93.3)	24 (51.1)	
No	1 (6.7)	23 (48.9)	
**Smoker, *n* (%)**			0.087
Yes	0 (0)	8 (17.0)	
No	15 (100)	39 (83.0)	
**Hormonal therapy, *n* (%)**			0.090
Yes	5 (33.3)	13 (27.7)	
No	10 (66.7)	34 (72.3)	

^a^ The results are presented as mean and standard deviation; mean (SD). ^b^
*p*-values are based on the Mann–Whitney non-parametric *U* test. Statistically significant results are indicated in bold.

**Table 2 ijms-25-10141-t002:** Disease characteristics.

	Endometriosis *n* = 47 ^a^
**Endometriosis diagnosis, years ago**	
Mean (SD)	2.8 (2.52)
Min, Max	1, 14
**Dysmenorrhea complaints, *n* (%)**	44 (93.6)
**Dysmenorrhea severity, *n* (%)**	
None/Mild/Moderate (<7)	11 (23.4)
Severe (≥7)	36 (76.6)
**Dyspareunia complaints, *n* (%)**	28 (59.6)
**Dyschezia complaints, *n* (%)**	16 (34.0)
**Rectorrhagia *n* (%)**	4 (8.5)
**Chronic pelvic pain, *n* (%)**	19 (40.4)
**Bowel symptoms, *n* (%)**	15 (31.9)
**Urinary tract symptoms, *n* (%)**	5 (10.6)
**Extra-pelvic endometriosis, *n* (%)**	2 (4.3)

^a^ The results are presented as mean and standard deviation; mean (SD).

## Data Availability

The data presented in this study are available upon request from the corresponding author. The data are not publicly available due to privacy restrictions.

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
