# Peer review of "The Role of NK and T Cells in Endometriosis"

_ijms, 2024, doi:10.3390/ijms251810141_

Round 1

Reviewer 1 Report

Comments and Suggestions for Authors

This is a very nice approach and can provide value to the endometriosis community. That said, I have some requests for clarity and improvement.

Major

- Treatment of race: First, be consistent. It's either Black or African (I recommend the former) and white or Caucasian. Next, be careful when describing race. You state it is demographic; however, it is unclear and unlikely that your groups are homogeneous. Race is not as clear an indicator of ancestry as we think and we are not good at determining race of others. If self identified it may be better as a sociopolitical. Lastly, and most concerning, is the lack of discussion on the extremely low # of black subjects in the study. There are no controls, so you cannot be positive that there aren't initial differences. With 7 vs 40 it is also hard to determine if this is real or just part of the natural variation. What is your power?

-Section 2.3 and 2.4. There are no graphs, tables, or actual values presented for this data despite you highlighting it early on in the discussion. It is of interest, but you need to provide the means and standard deviations as a minimum, not just the p-values. Especially when you are discussing 'trends' (which really means nothing as it isn't statistically different)

References: Some of your citations for studies showing differences in NK cells are review articles.  There are also a few recent articles on NT cells and endometriosis that are not cited that support your work.

Minor

- methods: was race self identified?

Reviewer 2 Report

Comments and Suggestions for Authors

In the present research authors collected peritoneal fluid (PF) and peripheral blood (PB) samples of patients affected by endometriosis to demonstrate differences in immunological profiles.

Specifically, differences in specific lymphocyte subsets (NK and T cells) were observed, suggesting potential immunological mechanisms in the evolution and clinical presentation of endometriosis.

The manuscript is well writtenn and data are clearly presented. To improve the overall quality i have a few suggestions:

1) Histolopathological data of surgically resected samples should be included to correlate blood samples findingsw with the immune infiltrate within tissue samples

2)Endometrioisis and cancer: discuss this association (refer to PMID: 32570258)  and clarify if studied samples were all benign or if some of them showed malignant features (atypical hyperplasia, endometrioid/clear cell carcinoma)

Round 2

Reviewer 1 Report

Comments and Suggestions for Authors

The authors have addressed my concerns.